CC U

5

# Seasonal variability of dissolved organic matter in the Columbia River: In situ sensors elucidate biogeochemical and molecular analyses

Urban Johannes Wünsch<sup>1,2</sup>, Boris Peter Koch<sup>2,1</sup>, Matthias Witt<sup>3</sup>, Joseph Andrew Needoba<sup>4</sup>

<sup>1</sup> University of Applied Sciences Bremerhaven, An der Karlstadt 8, D-27568 Bremerhaven, Germany

<sup>2</sup> Alfred-Wegener-Institut Helmholtz-Zentrum für Polar- und Meeresforschung, Am Handelshafen 12, D-27570 Bremerhaven, Germany

<sup>3</sup> Bruker Daltonik GmbH, Fahrenheitstraße 4, D-28359 Bremen, Germany

<sup>4</sup> Oregon Health and Science University Institute of Environmental Health, 3181 Southwest Sam Jackson Park Road,
 Portland, Oregon 97239-3098, United States of America

Correspondence to: Urban J. Wünsch (urbw@aqua.dtu.dk)

**Abstract.** The in situ detection of fluorescent dissolved organic matter (FDOM) at high temporal resolution is a powerful proxy to follow dissolved organic matter (DOM) dynamics and DOM flux to coastal oceans when FDOM measurements and dissolved organic carbon (DOC) are highly correlated. Here, we investigated the relationship between FDOM sensors and

- 15 DOC concentration in the lower Columbia River, USA in spring and summer 2013. Furthermore, we studied the seasonal variability of FDOM and chromophoric DOM (CDOM) optical indices, as well as the seasonal and spatial variability for the molecular characteristics of DOM using ultrahigh resolution electrospray ionization Fourier transform ion cyclotron resonance mass spectrometry (FT-ICR-MS). The fieldwork was conducted concurrently with the operation of in situ sensor platforms that recorded physical and biogeochemical data at hourly intervals. In situ FDOM and DOC concentration was
- 20 highly correlated and the relationship was used to quantify the river's DOC flux between March and August 2013. The average flux was 0.93 Gg d<sup>-1</sup>, which included over three-fold temporal variability (0.45 to 1.62 Gg d<sup>-1</sup>) associated with seasonal biogeochemical variability. Spectrofluorometry measurements demonstrated that FDOM parameters correlated with major seasonal biogeochemical shifts in the river associated with phytoplankton blooms and river discharge and thus revealed predictable seasonal patterns in DOM quality. FT-ICR-MS analyses elucidated these shifts on the molecular level:
- the relative abundance of 561 formulas, most of which contained N and S, correlated significantly with chlorophyll a, while 417 formulas (mostly CHO) correlated with CDOM absorbance at 254 nm.

## Keywords

Colored dissolved organic matter, Fluorescent dissolved organic matter, CDOM, FDOM, dissolved organic carbon flux, ultrahigh resolution mass spectrometry, FT-ICR-MS

#### **1** Introduction

Riverine organic matter is a large source of organic carbon to the oceans (approximately  $0.37 \times 10^{15}$  g C yr<sup>-1</sup>, Schlesinger and Melack 1981). Globally, riverine dissolved organic carbon (DOC) supplies the largest quantity of reduced carbon (0.25 x  $10^{15}$  g C yr<sup>-1</sup>), whereas the flux from individual rivers can vary by several orders of magnitude, depending on watershed characteristics (Hansell and Carlson, 2002; Spencer et al., 2013).

- Chromophoric, or colored, dissolved organic matter (CDOM) is the light absorbing fraction of DOM and its properties have
  been used as indicators of the chemical composition of DOM with regards to molecular weight and aromaticity (Helms et al. 2008; Weishaar et al. 2003). CDOM typically includes a portion that release photons after absorbance of light as
  fluorescence (FDOM) and this portion of DOM has been widely studied using spectrofluorometry techniques. Important
- biogeochemical processes are also reflected in changes in CDOM and FDOM properties; these include allochthonous vs. autochthonous carbon sources, and microbial degradation (Cory and McKnight, 2005; Parlanti et al., 2000). Furthermore, DOM absorbance and fluorescence facilitate in situ measurements using submerged sensors (Comeau et al., 2007; Gardner et al., 2005; Jannasch et al., 2008). Recently, the relationship between CDOM and DOC has been characterized for a large number of rivers in order to improve estimates of terrestrial carbon fluxes at the continental scale (Spencer et al., 2012;
- Stedmon et al., 2011).

With its high mass resolving power (> 400,000; i.e. ratio of ion mass to smallest separable ion mass difference), ultrahigh resolution Fourier transform ion cyclotron resonance mass spectrometry (FT-ICR-MS) is a powerful analytical method for resolving single mass formulas in the complex mixture of DOM (Stenson et al., 2003). The application of this technique has led to an increased understanding of the processing of DOM in the ocean (e.g. Koch et al., 2014; Kujawinski et al., 2004;

- Lechtenfeld et al., 2014) and in estuaries (e.g. Lechtenfeld et al., 2013; Sleighter and Hatcher, 2008). Other studies linked optical and chemical DOM properties (Kellerman et al., 2015; Stubbins et al., 2014).
  With an annual freshwater discharge of 7,790 m<sup>3</sup> s<sup>-1</sup>, and an annual DOC flux of 4.04 x 10<sup>11</sup> g, the Columbia River is one of the major North American sources of DOC to the Pacific Ocean (Hinck et al., 2006; Spencer et al., 2013). Hopkinson et al. (1998) compared DOM of numerous U.S rivers and concluded that Columbia River has a higher contribution of aliphatic
- DOM, which resulted in increased organic carbon bioavailability (Hopkinson et al., 1998). Further characterization of the chemical composition and seasonal dynamics of this carbon flux are needed to better evaluate the ultimate fate of this DOC source in the estuary and the coastal ocean. Since 2009 sensors deployed in the lower Columbia River estuary have measured FDOM along with a suite of other biogeochemical parameters (Baptista et al., 2015) that provide unprecedented spatial and temporal resolution for this large ecosystem (e.g. Gilbert et al. (2013)).
- However, the correlation between CDOM and DOC has previously been described as weak in comparison to other U.S. rivers (Spencer et al., 2012) and therefore the goal of this research was to improve our understanding of the correlations between FDOM and DOC in the context of variability in related biogeochemical cycles and hydrologic conditions. Our specific goal was to investigate if the existing high resolution in situ sensors can be used to quantify the seasonal variability

of the DOC flux to the coastal zone. Moreover, to better understand the influence of biogeochemical events (i.e. phytoplankton blooms and the spring freshet) on the composition of DOM we examined spectrofluorometric properties and used FT-ICR-MS to further characterize Columbia River DOC on the molecular level.

#### 2 Materials & Methods

#### 2.1 Study area and sampling

Samples for laboratory measurements were collected between 26 March and 20 August 2013 at two locations in the lower Columbia River (Fig. 1) that are sites of biogeochemical sensor platforms operated by the Center for Coastal Margin Observation and Prediction (Baptista et al., 2015). With a total of 61 samples (including replicates), station SATURN-08

- Observation and Prediction (Baptista et al., 2015). With a total of 61 samples (including replicates), station SATURN-08 (river mile 122) was the focus of our sampling campaign due to its accessibility (marina breakwater). Sampling at SATURN-08 included three time series efforts where hourly samples were collected (16 April 17 April, 29 May 30 May, 15 July 16 July). Due to limited access by boat, SATURN-05 (river mile 53) was sampled less frequent (n = 4), and primarily served as a site for the downstream comparison of molecular properties of DOM.
- 1 L amber glass bottles with PTFE-lined caps (Thermo Fisher Scientific, cleaned with 10 % HCl for 24 h) were filled with sample water, and stored at 4 °C. Filtration was performed immediately after return to the laboratory, using a 1 L glass filtration unit (Merck Millipore) and GF/F filters (Whatman, 47 mm, pre-combusted at 450 °C for 5 h). To create the necessary vacuum (< 200 mbar), a hand operated vacuum pump (Thermo Fisher Scientific) was used. Since combusted GF/F-filters can adsorb organic compounds (Maske and Garcia-Mendoza, 1994), the filtration unit was pre-rinsed with 200–</p>
- 300 mL of sample to saturate the filter, and rinse the system. Between samples, the filtration unit was thoroughly rinsed with ultrapure water (MilliQ<sup>®</sup> Plus 185<sup>®</sup> water purification system, conductivity 18.2 MΩ, total organic carbon < 5 pbb, 22 29 °C) before applying the next sample.

DOC samples were stored in 60 mL HDPE bottles at -20 °C; filtrates for spectroscopic measurements were maintained at 4 °C in 60 mL amber glass bottles (VWR International, pre-combusted at 500 °C for 5 h). The filtrate used for solid-phase

extraction was immediately acidified (pH 2) using hydrochloric acid (Suprapur grade<sup>®</sup>, Merck), and stored at 4 °C until further processing.

#### 2.2 Solid-phase extraction of dissolved organic matter

On four occasions, replicate solid-phase extractions were carried out at both sampling sites (n = 8,  $n_{total}$  = 16, Tab. 1). Solidphase extraction (SPE) was performed using PPL cartridges (200 mg, Agilent Technologies; (Dittmar et al., 2008)). Four extractions were carried out simultaneously using a Supelco VisiPrep<sup>®</sup> unit (Sigma Aldrich). Cartridges were equilibrated with 6 mL of methanol (LC-MS grade, Sigma Aldrich) and 6 mL of acidified ultrapure water (pH 2, hydrochloric acid, Suprapur<sup>®</sup> grade, Merck). 300 mL of sample filtrate was processed at a flow rate of 2.5 mL min<sup>-1</sup> using a peristaltic pump. The cartridges were rinsed with 6 mL of ultrapure water (pH 2) to remove salts, followed by drying with nitrogen (purity grade 5.0). The cartridges were stored at – 20°C until elution with ~1.4 mL methanol (LiChrosolv<sup>®</sup> grade, Merck). The

30 precise extraction volume was determined by weight.

5

# Biogeosciences Discussions

#### 2.3 Absorbance spectroscopy

UV/Vis spectra were obtained using a spectrophotometer (Tidas I, J&M Analytik AG) equipped with both Tungsten and Deuterium light sources (DH-2000-S, World Precision Instruments). Absorbance through a 1 cm quartz cuvette was scanned in the range of 200 – 723 nm. Ultrapure water was used as a reference; electronic noise was subtracted by measuring signal intensities with the light shutter closed. The mean signal intensity above 700 nm was subtracted from sample absorbance to account for instrument offsets due to particle scattering and differences in refractory indices (Green and Blough, 1994). Absorbance units were converted to napherian absorbance coefficients as follows:

$$a = \frac{2.303 * A}{l}$$
 (Eq. 1)

where a was the napherian absorption coefficient, A the measured absorbance, and l the measurement path length. The

10 specific ultraviolet absorbance (SUVA) was determined by dividing the UV absorbance at 254 nm (in  $m^{-1}$ ) by the DOC concentration measured in mg L<sup>-1</sup> (Weishaar et al., 2003).

#### 2.4 Fluorescence spectroscopy

Filtered samples were analyzed within 5 days after filtration. Spectra were obtained with a spectrofluorometer equipped with a twin channel beam (Fluoromax 4, Horiba Jobin Yvon). Emission of fluorescence was measured at 300–600 nm
(increments of 2nm) with excitation wavelengths of 240 – 450nm (increments of 10 nm). Sample temperatures were adjusted to 20°C during analysis. Instrument-specific correction factors were applied using manufacture supplied correction files. The validity of the instrument calibration was checked on a daily basis using the water Raman peak at excitation / emission 350 / 397 and did not vary more than ± 0.5 nm. The optical immaculacy of the quartz cuvette was checked by ensuring a flat emission profile of ultrapure water at an excitation of 280 nm. A daily blank (ultrapure water) was used to measure the Raman peak area at 350 nm for signal normalization.

To obtain comparable fluorescence spectra, also called excitation-emission matrices (EEM), the raw data was subject to post-processing. Observed fluorescence values were corrected for the inner filter effect by applying a correction factor (Parker, 1968):

$$F_{cor} = F_{obs} * 10^{OD_{ex} + OD_{Em}}$$
(Eq. 2)

- 25 where  $F_{obs}$ , and  $F_{cor}$  were the raw and corrected fluorescence. The necessary absorption values for the correction ( $OD_{ex}$  and  $OD_{em}$ ) were obtained from the corresponding UV/Vis spectrum. A daily blank (ultrapure water) was subtracted, and fluorescence counts were normalized to the Raman peak area. The fluorescence index (FI) was calculated as the ratio of emission signals at 470 nm and 520 nm obtained at 370 nm excitation (Cory and McKnight, 2005). The apparent fluorescence quantum yield at 350 nm was investigated by dividing the fluorescence emission integral from 300 to 600 nm at
- 30 an excitation of 350 nm by absorbance at 350 nm (procedure as described previously (Wünsch et al., 2015; Parker and Rees, 1960)). However, since our spectrofluorometer was not calibrated using e.g. quinine sulfate, we were unable to report quantum yield percentages. The biological index (BIX, an index indicating a recent contribution of autochthonous material)

was calculated as the ratio of fluorescence emission at 380 nm and 430 nm at an excitation of 310 nm (Huguet et al., 2009). The Humification Index (HIX) was calculated with the Raman normalized fluorescence emissions *I* at 254 nm excitation, as follows (Ohno, 2002):

$$HIX = \frac{\sum I_{435-480}}{\sum I_{300-345} + \sum I_{435-480}}$$
(Eq. 3)

5 Since the dataset resolution of EEMs differed from that necessary to calculate HIX, a linear interpolation was performed to obtain the necessary excitation and emission values. HIX values range from 0-1, and indicate the degree of DOM humification.

#### 2.5 Dissolved organic carbon measurement

- DOC was measured using high temperature catalytic oxidation. Organic carbon was oxidized to carbon dioxide, then 10 measured with a non-dispersive infrared detector (TOC/V<sub>CPN</sub>, Shimadzu). A 6.5 mL sample was poured into a precombusted glass vial, then acidified (30 μL 1 mol L<sup>-1</sup> hydrochloric acid, *pro analysi*, Merck) in an autosampler (ASI-V, Shimadzu). Acidified samples were sparged with oxygen to remove inorganic carbon. 50 μL acidified samples were injected onto the catalyst (heated to 680 °C) using the instrument's autosampler unit. DOC concentrations were averaged over three measurements.
- 15 The instrument was calibrated using laboratory standards ( $20 240 \mu mol L^{-1}$ , potassium hydrogen phthalate, and potassium nitrate (Wako Pure Chemical Industries). For quality control, the deep sea reference (DSR, Hansell laboratory, Miami) was used. The instruments detection limit was 7  $\mu mol L^{-1}$ ; DOC was measured with an accuracy of  $\pm$  5 % (Schmidt et al., 2009). The DOC concentration in solid-phase extracts was determined by drying 50  $\mu$ L of each methanol extract at 40°C with N<sub>2</sub>, and subsequent redissolving in ultrapure water.

#### 20 2.6 In situ sensor

30

At SATURN-08, a Satlantic LOBO platform (Jannasch et al., 2008) was operated by the Center for Coastal Margin Observation and Prediction (Portland, Oregon). The in situ data from the sensor package was obtained from instruments that included an optical nitrate sensor (Satlantic SUNA; (Johnson and Coletti, 2002)); a physical-chemical sensor measuring conductivity, temperature, pressure, dissolved oxygen, chlorophyll a, and turbidity (WET Labs Inc. WQM; (Orrico et al.,

25 2007)); a sensor for FDOM (WET Labs ECO); and a wet chemistry instrument for phosphate measurements (WET Labs Cycle Phosphate Sensor; (Barnard et al., 2009)). Sensors were cleaned and maintained monthly, following manufacturerrecommended protocols.

Periodic discreet samples were collected as a quality control measure for chlorophyll a and dissolved nutrients. Acid-washed 1 L Nalgene bottles were rinsed three times with sample water, then filled and transported back to the laboratory where they were analyzed using standard protocols (Welschmeyer 1994, Gilbert et al, 2013). No corrections were made to the in situ

nitrate and phosphate measurements, however chlorophyll a was adjusted using linear regression to account for the disparity

between the in situ measurement and the laboratory measurement according to the relationship  $y = 3.55 \text{ x} - 0.181 (R^2 = 0.93, n = 13).$ 

The in situ fluorescence was measured at an excitation wavelength of 370 nm and an emission wavelength of 460 nm. Since sensor electronics are affected by temperature and the emission of fluorescence is temperature dependent, a temperature correction was performed as described in (Watras et al., 2011). The sensor was placed in a dark PVC flow-through cell and submerged in 4.5 mg/L Suwannee River natural organic matter (Lot: 2R101N) at temperatures ranging from 4 to 35 °C. The temperature correction coefficient ρ was found to be -0.0297°C<sup>-1</sup>.

### 2.7 Hydrological and metrological data

River discharge at Bonneville Dam (river mile 146) and Beaver army terminal (river mile 53) was measured and provided by the US Army Corps of Engineers and the U.S. Geological Survey National Streamflow Information Program, respectively. The data was accessed through the CMOP online data access interface (www.stccmop.org/datamart/observation\_network/dataexplorer).

The meteorological data was retrieved from the United States National Climatic Data Center (http://www.ncdc.noaa.gov/).

- Data for SATURN-08 was collected at station GHCND:USC00358634 (Troutdale, OR, 45.5533 °N, -122.3886 °W). The 30year climate normal data (1981 - 2010) was obtained from station GHCND:USW00024229 (Portland International Airport, OR, 45.5958 °N, -122.6093 °W). Snowfall data was retrieved from station COOP:353402 (Government Camp, OR, 45.3014 °N, -121.74167 °W, product "TSNW - Total snow fall"). It should be noted that these three weather stations are insufficient to describe precipitation and snowfall events of the whole Columbia River watershed. However, for the purposes
- of this study, it was assumed that they provide a suitable indication of lateral inputs to the river at the sampling stations.

## 2.8 Fourier transform ion-cyclotron resonance mass spectrometry

For FT-ICR-MS analysis, DOM extracts were adjusted to an identical concentration of 4,850  $\mu$ mol L<sup>-1</sup> by dilution with methanol (LiChrosolv<sup>®</sup>, Merck). Samples were ionized with electrospray ionization (Apollo II electrospray source, Bruker Daltonik GmbH) at an infusion flow rate of 2  $\mu$ L min<sup>-1</sup> in negative ion mode.

Analyses were performed on a Bruker SolariX XR FT ICR-MS (Bruker Daltonik GmbH) equipped a ParacellTM and a 12 T-refrigerated, actively shielded superconducting magnet (Bruker Biospin).

Initial mass spectra calibration was performed with arginine clusters using a linear calibration. Ion accumulation time was set to 0.03 s, and 200 scans were added to one mass spectrum ranging from m/z (mass to charge ratio) 100 - 1,000. Mass spectra were acquired with 8 Megawords data points, resulting in a resolving power of 790,519 at the highest peak (m/z 381.11932).

After analysis, mass spectra were recalibrated internally with a set of marine DOM molecules (Koch et al., 2014). Molecular formulas were calculated by considering the following isotopes:  ${}^{1}H_{0-120}$ ,  ${}^{12}C_{0-50}$ ,  ${}^{13}C_{0-1}$ ,  ${}^{16}O_{0-35}$ ,  ${}^{14}N_{0-2}$ ,  ${}^{32}S_{0-1}$ , and  ${}^{34}S_{0-1}$  (m/z range of 200 - 600). Formulas were restricted to integer double bond equivalent (DBE) values (Stenson et al., 2003); only

10

15

formulas with a mass tolerance of  $\pm 0.5$  ppm were considered. Peak intensities were normalized to the highest ion peak in each spectrum, and the dataset was restricted to a relative peak intensity of  $\geq 0.3$  %. Potential anthropogenic surfactants that are listed in a surfactant database (www.terrabase-inc.com), and other contaminations (detected in the process blanks) were removed. The dataset was restricted to formulas with abundances in respective process duplicates, i.e. a formula was

5 disregarded if only detected in one of two replicates. This approach yielded up to 4,286 formulas per sample. It should be considered that every assigned molecular formula most likely implies an immense structural diversity of isomers (Hertkorn et al., 2008). Since FT-ICR-MS alone is unable to distinguish between such structural isomers, we cannot draw conclusions as to the extent to which this applies to an individual molecular formula.

Intensity-weighted average (wa) molecular masses and element ratios were calculated from the normalized peak magnitudes. The DBE for identified molecular formulas was calculated with the number of atoms of an element i ( $N_j$ ) and its valence  $V_j$ 

as follows (Koch et al., 2005):

$$DBE = 1 + \frac{\int_{i}^{imax} N_{i} * (V_{i} - 2)}{2}$$
(Eq. 5)

The relative peak intensities of ubiquitous formulas were correlated with various parameters (in situ chlorophyll a, DOC, and in situ FDOM), using the Pearson product-moment correlation coefficient (r) to obtain information about the quality and orientation of dependencies. The significance threshold was p 

#### **3 Results**

#### 3.1 Seasonal variability of biogeochemical parameters

During the sampling campaign in 2013 (Fig. 2, data with grey background), daily maximum water temperatures ranged from 5.3 °C at the beginning of March to 22.9 °C at the end of August. The total precipitation between March and August 2013

was 257.2 mm. During the winter season (October 2012 – September 2013), the total snowfall at Government camp (station COOP:353402) was 5.7 x 10<sup>3</sup> mm (data not shown).

A seasonal bloom of phytoplankton was observed as a peak in chlorophyll a during late March 2013 (Fig. 2c). Nitrate was elevated in early spring (until 13 April 2013), and steadily decreased from 30  $\mu$ mol L<sup>-1</sup> to 6.5  $\mu$ mol L<sup>-1</sup> by 22 May 2013. Phosphate concentrations were below the sensor detection limit (0.075  $\mu$ mol L<sup>-1</sup>) from March until mid-May 2013, and never

- exceeded 0.3  $\mu$ mol L<sup>-1</sup> for more than one day (data not shown).
- The seasonal discharge maximum at SATURN-08 was observed on 15 May 2013, which was followed by the seasonal FDOM maximum on 27 May 2013. With 1000 m<sup>3</sup> s<sup>-1</sup> at SATURN-08, the seasonal discharge maximum was comparatively low (Fig. 3a). Moreover, spring and summer 2013 were characterized by multiple smaller discharge peaks, whereas other years (e.g. 2011) only show one distinct peak associated with the spring freshet (Fig. 3a). However, the relative discharge
- contribution of the Willamette River and other regional tributaries to the Columbia River in 2013 followed the typical patterns and magnitudes: With an average of around 15 %, the smallest contribution by volume was observed during July 2013, while reaching up to 65 % during November 2013 (Fig. 3b).

Many of the biogeochemical parameters were comparable at both sensor stations between 2012 and 2013. For example, the seasonal trends of water temperatures were almost identical (Fig. 2d). Moreover, chlorophyll a levels were also comparable

with the notable exception of higher levels between May and November 2013, including a small summer phytoplankton bloom during June 2013 at SATURN-05, which was not observed to the same extend at SATURN-08. Discharge was consistently higher at SATURN-05. Water discharge, along with water turbidity during early winter 2012 was proportionally more elevated at the downstream station SATURN-05.

In situ FDOM was a notable exception to the otherwise high similarity between biogeochemical data at both stations.

Although showing similar responses to short-term events, e.g. during times of high organic matter concentrations in the river, the instrument at both stations showed different long-term trends. For example, between November 2012 and August 2013, the FDOM sensor at SATURN-05 indicated a noticeable overall decrease in that period, while the signal returned to an apparently constant baseline value at SATURN-08.

#### 3.2 Optical properties of dissolved organic matter

To judge the extent of the seasonal variability of CDOM and FDOM properties, the analysis was focused on samples from station SATURN-08, since the dataset was better suited to investigate correlations (n = 61). The highest relative variability was observed in the DOM aromaticity indicator SUVA<sub>254</sub>, while the FI had the lowest variability (Table 2). Correlation

analyses were conducted to relate the variability of the optical indices to the biogeochemistry of the Columbia River (Fig. 4). With the exception of the SUVA<sub>254</sub> data, the investigated parameters were normally distributed (Shapiro-Wilk normality test p < 0.001). Therefore, Spearman's rank correlation was used to conduct analyses of the relationships between parameters. Seasonal changes in the EEM-derived FI correlated with chlorophyll a (Fig. 4a). BIX, HIX and SUVA<sub>254</sub> correlated with

5 DOC (Figs. 4b, c, and d). The apparent fluorescence quantum yield did not covary with any of the recorded biogeochemical parameters (data not shown).

The dataset for the linear regression of DOC and FDOM was not normally distributed (Shapiro-Wilk normality test, p < 0.001). However, to judge the linearity of relationships, a parametric correlation analysis was conducted and DOC was found to be correlated to in situ FDOM ( $R^2 = 0.95$ , p < 0.001, Fig. 5). Moreover, in situ fluorescence and FDOM

fluorescence of filtered samples at excitation / emission 370/460 nm correlated significantly (R<sup>2</sup> = 0.95, p < 0.001, Fig. 5). It 10 was therefore assumed that in situ fluorescence was sufficiently representative of FDOM fluorescence at the same wavelengths.

As a result of these findings, we used the linear absorbance model to predict DOC using the in situ FDOM data. The average model uncertainty (accounting for uncertainty in the estimated coefficients, and the variance in observations) was  $\pm 11.4 \ \mu mol \ L^{-1} \ or \pm 8.75 \ \%.$ 

15

#### 3.3 Molecular characteristics of dissolved organic matter

The recorded FT-ICR mass spectra displayed very similar distributions of peak magnitudes across samples (Fig. 6a): The highest abundances were observed near m/z 400, in the center of the van Krevelen plot (Fig 6b). On average, we assigned  $3994 \pm 222$  molecular formulas in each sample. The average molecular characteristics for all samples are presented in Table

20 3.

> On a presence or absence basis, the molecular analyses indicated a high degree of similarities between all samples. 3922 out of a total of 5122 molecular formulas were detected at both stations, and 59.4 % (3044) were found in all samples. 85.0 % (3671) and 70.8 % (3345) of the identified molecular formulas were ubiquitous at stations SATURN-05 and SATURN-08, respectively.

- However, the dataset also contained unique molecular formulas (444, or 0.8 % at station SATURN-08, Fig. 7a, and 382, or 25 0.7 % at station SATURN-05, Fig. 7b). At SATURN-08, samples from the spring phytoplankton bloom showed 205 unique molecular formulas that were low in oxygen content and highly saturated (O/C <0.5 and H/C >1.2). However, at SATURN-05, the 90 unique molecular formulas did not show a similar characteristic cluster and were evenly distributed in the compositional space. At both locations, the unique molecular formulas during the spring freshet were dominated by low H/C,
- but high O/C ratios. Interestingly, samples at SATURN-05 showed high amounts of unique molecular formulas during 30 August 2013 with low O/C and high H/C ratios; a trend that was not equally observed at the upriver station SATURN-08. Since the degree of similarity between stations and samples was far greater than the amount of unique molecular formulas, we further investigated molecular patterns using the ubiquitous molecular formulas. The degree of similarity between

samples, based on the 3044 ubiquitous molecular formulas and their normalized molecular formula abundances (molecular formula abundance divided by the maximum abundance at the respective stations) is shown in the left panel of Fig. 8. The cluster analyses were based on the seasonal abundance changes of each individual molecular formula. Each seasonal event is separated into individual clusters, while process replicates, with the exception of the spring bloom samples, formed their own

5 respective clusters.

To elucidate this cluster analysis, we investigated the average seasonal changes of ubiquitous molecular formula abundances for all sampling times (Fig. 8, right panel). We found a high degree of variability for molecular formulas that can be categorized by their location in the van Krevelen plot. The first group, with H/C (0.9 - 1.3) and O/C ratios (0.4 - 0.6), was high during April 2013 (Fig. 8 top row, right panel). Samples collected during the rain event displayed a broader range of

- 10 highly abundant molecular formulas extending to the outer margin of the van Krevelen plot. The second group, with low H/C (< 1.0) and high O/C ratios (> 0.5), had characteristically high relative peak magnitudes during June 2013, when the Columbia River's annual discharge maximum was reached. On the other hand, this group was found to be lower in abundance in all remaining samples. At the same time, most molecular formulas in the first group were found to be comparatively low during June (and also August).
- To elucidate the peak magnitude variability of ubiquitous molecular formulas (Fig. 8), we combined the in situ dataset at both stations with FT-ICR-MS spectra obtained from samples at this station, and carried out correlation analyses to establish molecular markers for biogeochemical events. Chlorophyll a levels, and the napherian absorbance coefficient at 254 nm were chosen as proxy for the phytoplankton bloom, and the increased organic matter export during the spring freshet, respectively. We found a total of 978 molecular formulas that satisfied the significance threshold (p 

#### **4** Discussion

The Columbia River is well-studied; previous investigations have documented annual patterns in springtime phytoplankton blooms, discharge maxima, and nutrient availability (Sullivan et al., 2001). Since 1995, the U.S. Geological Survey sampled the Columbia River on a monthly basis to collect long-term time series data, including CDOM absorbance (Spencer et al.,

5 2012). However, such constant monitoring relies on a relatively low sampling frequency that is, if at all, usually only supported by basic sensor products, such as river discharge and water temperature. In contrast, this study was supported by a more comprehensive array of in situ sensors, allowing for reactive sampling during different environmental conditions that captured the biogeochemical variability of DOM for optical and molecular studies.

The continuous data acquisition at both in situ stations was successfully used to identify biogeochemical events during the sampling campaign in spring and summer 2013. As evident by elevated chlorophyll concentrations, the annual spring bloom developed during late March and early April. The dominant phytoplankton species during that time was *Asterionella formosa* (Maier and Peterson, 2014). The timing of the spring bloom onset matched that observed in previous studies of the Columbia River (Sullivan et al., 2001), and was likely driven by light and nutrient availability. River discharge is known to be a contributing factor for the length and intensity of phytoplankton blooms in the Columbia River (Sullivan et al., 2001).

Correspondingly, during early spring 2013, rainfall during the peak of the bloom contributed to increasing river discharge, causing a steep decline in phytoplankton abundance.

Phosphate concentrations were low, compared to historical data, especially during early spring. During this time, historic data (1978-1994, USGS data at river kilometer 227) indicates phosphate concentrations of  $> 0.5 \mu mol L^{-1}$  (Sullivan et al., 2001), whereas during early spring 2013, phosphate was below our sensor's detection limit ( $< 0.075 \mu mol L^{-1}$ ). Given the

20 molar Redfield ratio of approximately 16N:1P (Redfield, 1934), this indicates that phytoplankton growth was probably phosphate-limited as nitrate to phosphate ratios were never found to be below 33.8. Nonetheless, despite very low phosphate we observed a freshwater diatom bloom at typical levels (approximately 40 µg chlorophyll a L<sup>-1</sup>), suggesting that, although low, phosphate was never fully depleted, rather, instantly utilized and rapidly recycled.

The seasonal pattern of nitrate was typical for a temperate environment, and concentrations matched previous studies in the Columbia River (Sullivan et al., 2001). High nitrate concentrations during winter were attributable to reduced nitrogen uptake, and increased terrestrial nitrate runoff (Wall et al., 1998). In contrast, increased primary productivity and low levels of terrestrial nitrate input during spring and summer were likely responsible for lower nitrate concentrations.

As apparent from the meteorological data, spring and summer 2013 had low amounts of rainfall (45 mm or 15 % less rain from March to August, compared to the 30-year climate normal), and less snowfall (25.6 % less, compared to the winter

season 2010/2011). This is directly reflected in the comparatively low discharge maximum at Bonneville dam (30 % lower in 2013 compared to 2011; Fig. 3a). Moreover, the existence of multiple comparable discharge peaks during spring, together with coinciding high amount of rainfall during May 2013 and the overall low snowfall during the winter of 2012 / 2013 suggests that the seasonal discharge maximum was not primarily driven by snowmelt. Rather it suggests that, although the

base of the elevated discharge during spring and summer was induced by snowmelt, the discharge peaks were caused by increased lateral runoff due to the rainfall events. Therefore, although the highest seasonal FDOM signals during late May and early June coincided with the discharge maximum, the FDOM peak most likely did not represent a typical snowmeltdominated spring freshet, but an increased contribution of terrestrial runoff to the river. The influence of such inter-annually

varying biogeochemical conditions on the organic matter characteristics of the Columbia River remains unknown, and 5 should thus be subject to further studies.

The comparison data products from both in situ sensors beyond the sampling campaign allowed insights into the variability within the Columbia River. As stated above, many parameters showed significant degrees of overlap (e.g. water temperature) between 2012 and 2013. However, some seasonal patterns also differed between both stations. For example, water discharge

- 10 and turbidity during the winter months of 2012 showed a disproportionate increase at the downstream location SATURN-05 (Fig. 2a and Fig. 3b). During this period of increased rainfall, the Willamette River receives more terrestrial runoff, which likely contributes to the increased discharge and increased nitrate concentrations at SATURN-05. Contrary to the Willamette watershed, precipitation upstream of SATURN-08 in the Columbia River watershed partially falls as snow and is retained behind multiple hydropower dams and therefore has less direct impact on river discharge.
- Interestingly, during July 2013, chlorophyll a levels indicate the presence of a low level phytoplankton bloom at the 15 downstream station SATURN-05, that was not observed upstream, at SATURN-08. During this time, SATURN-05 showed increased nitrate that originated from the Willamette River, as apparent from data of the USGS National Streamflow Information program (data not shown). Since the Willamette River is usually also higher in phosphate (Prahl et al., 1997), increased nutrient availability likely caused a second summer phytoplankton bloom at SATURN-05 that was not observed at
- SATURN-08. 20

Most notably, in situ FDOM differs considerably between SATURN-05 and SATURN-08. Since a good agreement between in situ FDOM and DOC was found at SATURN-08, the quality of data acquired from SATURN-05 was further investigated. Unfortunately, due to the low number of samples (n = 8), a detailed analysis of the data validity using DOC measurements was difficult at SATURN-05. For the available data, no significant correlation between in situ FDOM readings and DOC

- 25 was found. This was also true when considering the extended dataset of in situ FDOM and DOC (measured by USGS) between 2009 and 2013. Therefore, the differences in readings between SATURN-05 and SATURN-08 are likely caused by one or several disturbances. Firstly, instrumental issues could have affected readings at SATURN-05. However, it has to be noted that samples discussed in Spencer et al. (2012) were taken at the same location and DOC was also found to be not well correlated to CDOM absorbance of filtered samples. Therefore, other factors, such as local point sources of industrial or
- 30 urban pollution could have significantly influenced the relationship on a local level since SATURN-05 is located downstream an area affected by industrial activities.