# Peer review of "Seasonal variability of dissolved organic matter in the Columbia River: In situ sensors elucidate biogeochemical and molecular analyses"

_Biogeosciences, 2016_

## Referee Comment (RC1) · Anonymous Referee #1 · 10 Aug 2016

General Comments: In this manuscript the authors examined data such as FDOM, chlorophyll fluorescence, temperature, nitrate, and turbidity measured with in situ sensor platforms deployed at two locations in the lower Columbia River, USA combined with discrete measurements of DOC concentrations, CDOM absorption and fluorescence in addition to molecular signatures using FT-ICR-MS during March –August 2013 (spring – summer). DOC fluxes for the sampling period were calculated based on the relationship between FDOM and DOC. Relationships between DOC and CDOM/FDOM optical indices (HIX, BIX, SUVA) were also investigated. Furthermore, molecular characteristics were examined for the spring events (phytoplankton bloom, rain event,

freshet) and for the summer sampling period. While the overall measurement approach was good, the results and the interpretations of the results were weak and sometimes speculative. Many papers that were cited were missing from the reference list.

Specific comments: Abstract, lines22-24: "....FDOM parameters correlated with major seasonal biogeochemical shifts in the river associated with phytoplankton blooms and river discharge and thus revealed predictable seasonal patterns in DOM quality." The results do not support this conclusion. Abstract, lines 25-26: This conclusion also not supported by the results - very speculative.

Page 2, line 5: Spencer et al. 2013 should be Spencer et al. 2012.Page 2, line 8: Weishaar et al. 2003 (missing reference) Page 5, line 7: "naphierian" should be Napierian Page 5, line 20: (Parker, 1968) missing reference Page 5, line 30: (Wunsch et al. 2015; Parker and Rees 1960) missing references Page 7, line 6: (Watras et al. 2011) missing reference 2.7 Hydrological and metrological data should be "Hydrological and meteorological data" Page 8, line 13: "The relative peak intensities of ubiquitous formulas were correlated with various parameters...." These results are not shown in the manuscript Page 9, line 12: "With 1000 m3 s-1 at SATURN-08, the seasonal discharge maximum was comparatively low (Fig. 3a)." The discharge is more like 10,000 m3 s-1. This statement needs to be corrected. Page 9, lines 31-32: "The highest relative variability was observed in the DOM aromaticity indicator SUVA254, while the FI had the lowest variability.." The implications of using these optical indicators remains unclear. This is briefly addressed in the discussion section (Page 15) with the conclusion that DOM was clearly terrestrially dominated (which is expected) and low in authochthonous DOM. It would help if the authors better explain the need for using these indices and the observed variability. Page 11, line 6: "To elucidate this cluster analysis, we investigated the average seasonal changes of ubiquitous molecular formula abundances for all sampling times (Fig. 8, right panel)." It appears that the changes investigated were linked to events (spring bloom, spring freshet, spring rain event) rather than a seasonal study. Page 12, line 15: Correspondingly, during

early spring 2013, rainfall during the peak of the bloom contributed to increasing river discharge, causing a steep decline in phytoplankton abundance" This is not evident from Fig. 2. Pages 15-16, lines 33, 1: ...The high abundance of such molecular formulas during the spring freshet could therefore explain shifts in SUVA254, BIX, and HIX." These shifts are not evident in any of the figures -too speculative. Page 16, lines 13-14: "Shifts in the fluorescence index indicated increased levels of fresh microbial DOM, demonstrating the link between primary and secondary production." These shifts are not evident in the figures or tables and the link between primary and secondary production too speculative.

Figures: Fig. 2(b): It is not clear why a decreasing FDOM trend is observed at SATURN-05 between ~5/12 to ~8/13. Fig. 5: If data points for May and June associated with the spring freshet are excluded it does not appear that DOC and FDOM are well correlated. Also, in Spencer et al. 2012, Columbia River exhibited weak relationship between CDOM absorption and DOC likely due to significant impoundment of waters within its watershed. This factor is rather complex but should be considered in this study. Fig.7: Not clear what is being presented here. These are at two different stations (SATURN-08 and SATURN-05) but at least visually two plots look the same. Were the data from the two stations combined?

---

## Referee Comment (RC2) · Anonymous Referee #2 · 19 Aug 2016

This is a nicely written paper describing seasonal variation of DOM and their molecular characteristics in the lower Columbia River. Such variation is mostly controlled by biogeochemistry and hydrology of the river. Using in-situ sensors to study biogeochemistry of river systems is advantageous over discrete sampling. This paper demonstrates there can be many new information and results to be found even in well studied river systems, such as the Columbia River. For example, using in-situ FDOM to quantify DOC and its flux is not new, but it is a new finding for Columbia River. I think this is encouraging even it is somewhat contradicting with what Spencer found in the past. The manuscript worth publication. I have a few comments, and hope they can

help the authors to make the paper stronger.

1. There was a significant difference between FDOM data between SATURN-08 (S8) and SATURN-05 (S5), and the relationship between FDOM and DOC only applies to S8. Although the authors gave some explanations in the paper, it may worth digging out more details, as the FDOM-DOC relationship is a major finding of the paper. The authors suggest that the difference of FDOM between the two stations might not be due to data quality, but rather local sources of changes down stream of S8. As they mentioned, there are historical data of DOC vs. FDOM in the lower Columbia River. I suggest they dig out those data and overlay their data to see if there is any consistency or inconsistency in all data and the relationship. If they believe there are something 'unusual' down stream of S8, can they try to use their available data to investigate the nature and effects of such sources? I think spending more effort into this may gain more insights into the FDOM and DOC relationship, making the conclusion stronger.

2. In a few places in the introduction and abstract, the text seems to suggest river discharge is one of biogeochemical factors (e.g., p3 at the beginning), which is not. It is hydrology. Need to be consistent throughout the paper.

3. p2, L2, the reference is still from 1981; I believe there should be newer update.

4. Eq 1. Please specify what is the purpose of calculating napherian absorbance.

5. p6, L21, what is LOBO?

6. Section 2.6, no performance metrics of the in-situ sensors were given. What are their precision/accuracy, data quality etc.? Need to be careful what the quality these sensors can deliver. They may not always give the data quality as suggested by the manufactures. Should more systematically describe how these sensors were used and calibrated, and how the data were quality controlled, and in what accuracy and precision these data can be trusted.

7. p6, L30, 'No corrections were made to the in-situ nitrate and phosphate measurements'. Why not, since there are discreet samples for nutrients measured?

8. p9, L18, 65%. It is more like 60% to me.

9. p10, L8, 'parametric correlation analysis'. What is this analysis exactly? Linear analysis?

10. p10, L8-13. The description here is confusing. Fig. 5 did not show Flu vs. FDOM relationship. What the two relationships here mean or imply then?

11. Between Section 4 and Section 4.1, the text here does not seem follow a logic way for presentation. May want to organize it into one or few sub-sections.

12. p12, L15, says rainfall caused decline in phytoplankton abundance. Any data or figure to show?

13. p12, L26, why there is increased terrestrial nitrate runoff in winter?

14. p12, L33, 'high' rainfall during May 2013. But it says low rainfall before this?

15. p14, L8, -7.57 vs. -4.7. I think the measurement error margin was about 10 umol/L, correct? So the difference is within the error margin, is it not? If yes, the conclusion here is not valid. How much of difference is in slopes here? Can that slope difference tell us some information?

16. Figs 2 and 4. There are no captions for sub plots (a), (b), . . .

17. Figure 10. Is there an explanation why USGS data are different than the modeled DOC?

---

## Author Comment (AC1) · 26 Aug 2016

We thank the anonymous reviewer for her / his constructive comments. The response below contains the reviewers comments (marked with "RC1: ") followed by our response (marked with "Response: "). Please note: As a consequence of this review, we will provide a revised version of the manuscript at a later time.

RC1: General Comments: In this manuscript the authors examined data such as FDOM, chlorophyll fluorescence, temperature, nitrate, and turbidity measured with in situ sensor platforms deployed at two locations in the lower Columbia River, USA

combined with discrete measurements of DOC concentrations, CDOM absorption and fluorescence in addition to molecular signatures using FT-ICR-MS during March – August 2013 (spring – summer). DOC fluxes for the sampling period were calculated based on the relationship between FDOM and DOC. Relationships between DOC and CDOM/FDOM optical indices (HIX, BIX, SUVA) were also investigated. Furthermore, molecular characteristics were examined for the spring events (phytoplankton bloom, rain event, freshet) and for the summer sampling period. While the overall measurement approach was good, the results and the interpretations of the results were weak and sometimes speculative. Many papers that were cited were missing from the reference list.

Response: The missing references were caused by an unfortunate bug in the citation management software. The list will be updated in the revised version of the manuscript.

RC1: Specific comments: Abstract, lines22-24: "....FDOM parameters correlated with major seasonal biogeochemical shifts in the river associated with phytoplankton blooms and river discharge and thus revealed predictable seasonal patterns in DOM quality." The results do not support this conclusion.

Response: Deleted "predictable" in the cited sentence. In our opinion, the remaining conclusion is supported by the results.

RC1: Abstract, lines 25-26: This conclusion also not supported by the results - very speculative.

Response: It is not clear which part of this conclusion is speculative. Lines 25-26 report results that are directly reflected in the FT-ICR-MS dataset. The conclusion "correlated significantly" was drawn by means of transparently described methods and datasets.

RC1: Page 2, line 5: Spencer et al. 2013 should be Spencer et al. 2012.

Response: The claim "flux from individual rivers can vary by several orders of magnitude, depending on watershed characteristics" is supported by table 1 in Spencer

2012, where annual discharge, watershed % wetland, as well as annual DOC yields are provided.

RC1: Page 2, line 8: Weishaar et al. 2003 (missing reference)

Response: See response to general comments. Reference will be provided in the revised manuscript.

RC1: Page 5, line 7: "naphierian" should be Napierian

Response: The spelling is corrected in the revised version of the manuscript

RC1: Page 5, line 20: (Parker, 1968) missing reference

Response: See response to general comments. Reference will be provided in the revised manuscript.

RC1: Page 5, line 30: (Wunsch et al. 2015; Parker and Rees 1960) missing references

Response: See response to general comments. Reference will be provided in the revised manuscript.

RC1: Page 7, line 6: (Watras et al. 2011) missing reference 2.7 Hydrological and metrological data should be "Hydrological and meteorological data"

Response: Reference added to the bibliography and spelling corrected.

RC1: Page 8, line 13: "The relative peak intensities of ubiquitous formulas were correlated with various parameters...." These results are not shown in the manuscript

Response: These results are shown in Fig. 9, mentioned in results section 3.3 (page 11), and discussed in the discussion section 4.2 (page 15).

RC1: Page 9, line 12: "With 1000 m3 s-1 at SATURN-08, the seasonal discharge maximum was comparatively low (Fig. 3a)." The discharge is more like 10,000 m3 s-1. This statement needs to be corrected.

Response: The statement is corrected in the revised version of the manuscript.

RC1: Page 9, lines 31-32: "The highest relative variability was observed in the DOM aromaticity indicator SUVA254, while the FI had the lowest variability.." The implications of using these optical indicators remains unclear. This is briefly addressed in the discussion section (Page 15) with the conclusion that DOM was clearly terrestrially dominated (which is expected) and low in authochthonous DOM. It would help if the authors better explain the need for using these indices and the observed variability.

Response: More detailed explanations for the interpretation of the optical indices were added to the methods section 2.4. We added another sentence explaining the need for optical indices in our study at the beginning of section 4.2 (page 15).

RC1: Page 11, line 6: "To elucidate this cluster analysis, we investigated the average seasonal changes of ubiquitous molecular formula abundances for all sampling times (Fig. 8, right panel)." It appears that the changes investigated were linked to events (spring bloom, spring freshet, spring rain event) rather than a seasonal study.

Response: The wording was changed to "changes between events", we recognize the concern of the reviewer.

RC1: Page 12, line 15: Correspondingly, during early spring 2013, rainfall during the peak of the bloom contributed to increasing river discharge, causing a steep decline in phytoplankton abundance" This is not evident from Fig. 2.

Response: The statement refers to Fig. 2(a) and (c), in particular the highlighted part(grey): Declining chlorophyll a abundance and increasing river discharge. This inverse relationship is hard to identify since the graph provides an overview of the entire season. Our statement is supported by the inverse correlation of chlorophyll a and river discharge during this time (April 1st – April 13th, $R2 = 0.76$). Moreover, as stated in this paragraph, earlier studies by Sullivan et al. showed similar findings. To support our conclusion, this correlation will be mentioned in the revised manuscript.

RC1: Pages 15-16, lines 33, 1: ...The high abundance of such molecular formulas during the spring freshet could therefore explain shifts in SUVA254, BIX, and HIX." These shifts are not evident in any of the figures -too speculative.

Response: As explained in the cited paragraph, we state that CDOM-correlated formulas had low H/C ratios and above average double bond equivalents. These two FT-ICR-MS metrics, as well as the evident correlation (all together summarized in Fig. 9) increase the likelihood that this particular part of the FT-ICR-MS dataset explains CDOM and FDOM variability (as the presence of an aromatic ring is a prerequisite for fluorescence of organic matter). In this context, we also referred to previously published work by Kellerman et al. which established similar relationships with a bigger dataset in Swedish lakes.

RC1: Page 16, lines 13-14: "Shifts in the fluorescence index indicated increased levels of fresh microbial DOM, demonstrating the link between primary and secondary production." These shifts are not evident in the figures or tables and the link between primary and secondary production too speculative.

Response: With chlorophyll a, the only available parameter to estimate biological activity concerned primary production. However, the fluorescence index (Parlanti et al.) has previously been established as an indicator for freshly produced microbial DOM (derived from the degradation of algae). As such, the shifts are not evident from figures or tables, but our conclusion can be drawn from the context of the results. To emphasize the speculative nature of our conclusion, "demonstrating the link" was substituted by "indicating a possible link".

RC1: Figures: Fig. 2(b): It is not clear why a decreasing FDOM trend is observed at SATURN-05 between _5/12 to _8/13.

Response: These data were discussed in section 4, page 13, lines 26-31. We concluded that the differences in readings between the two stations were caused by disturbances. The reviewer is correct in stating that it is not clear why the decrease between

May and August 2013 was observed. At this point, we have no possible explanation other than technical problems with the sensor. Since the sensor was recalibrated in the meantime, it is not possible to further investigate the issue.

RC1: Fig. 5: If data points for May and June associated with the spring freshet are excluded it does not appear that DOC and FDOM are well correlated. Also, in Spencer et al. 2012, Columbia River exhibited weak relationship between CDOM absorption and DOC likely due to significant impoundment of waters within its watershed. This factor is rather complex but should be considered in this study.

Response: The reviewer is correct in stating that the relationship between DOC and FDOM would not be strong if data from May 2013 were excluded. However, DOC values were highest during that period and to obtain a robust model of in situ data, these elevated readings are vital for several technical reasons concerning the accuracy of the in situ readings: (1) In situ readings are always affected by disturbances from particles that might vary between months, (2) readings are always affected by seasonal changes in temperature, and (3) unknown factors such as sunlight, diurnal variations in the power supply (solar power). While we corrected for (2) as mentioned in the manuscript, other factors ((1) and (3)) are difficult to investigate and therefore not quantifiable. It is therefore highly likely that these readings (along with the 5 % precision of the DOC measurement) caused the relatively weak relationship between DOC and FDOM when the overall variation was small. We added a paragraph in the discussion (revised manuscript): "However, if data from May 2013 were excluded, the relationship between DOC and FDOM would not be significant. This is likely caused by factors contributing to the measurement uncertainty, such as variations in water turbidity, imperfect temperature correction and slight variation in FDOM quality at the wavelength pair used by the sensor."

RC1: Fig.7: Not clear what is being presented here. These are at two different stations (SATURN-08 and SATURN-05) but at least visually two plots look the same. Were the data from the two stations combined?

Response: The plot highlights unique molecular formulas at station SATURN-08 (b) and SATURN-05 (b) against the background of all molecular formulas (grey, both stations combined). The similarity the reviewer refers to might arise from the grey background, or (as stated throughout the manuscript) the fact that SPE-DOM from both stations was relatively similar. The figure legend was adopted for clarification.

---

## Author Comment (AC2) · 8 Sep 2016

We would like to thank the anonymous reviewer #2 for her / his constructive comments that helped to improve the manuscript. The response below contains the reviewers comments (marked with "RC2: ") followed by our response (marked with "Response: "). Please note: As a consequence of this review, we will provide a revised version of the manuscript at a later time.

RC2: There was a significant difference between FDOM data between SATURN-08 (S8) and SATURN-05 (S5), and the relationship between FDOM and DOC only applies

to S8. Although the authors gave some explanations in the paper, it may worth digging out more details, as the FDOM-DOC relationship is a major finding of the paper. The authors suggest that the difference of FDOM between the two stations might not be due to data quality, but rather local sources of changes down stream of S8. As they mentioned, there are historical data of DOC vs. FDOM in the lower Columbia River. I suggest they dig out those data and overlay their data to see if there is any consistency or inconsistency in all data and the relationship. If they believe there are something 'unusual' down stream of S8, can they try to use their available data to investigate the nature and effects of such sources? I think spending more effort into this may gain more insights into the FDOM and DOC relationship, making the conclusion stronger.

Response: A comparison of the FDOM / DOC relationship between SATURN-05 and SATURN-08 is not possible, since no historic FDOM measurements are readily available from USGS to our knowledge. However, CDOM absorbance at 254 nm and DOC at SATURN-05 were sampled by USGS from 1999 onwards. As a result of the reviewers comment, we investigated the USGS dataset to further investigate and provide more detail. In total, 185 samples spanning the years 1999 to 2016 are available to date (as of 08/23/2016). The overall relationship between DOC and CDOM absorbance at 254 nm was significant, but weak ($R^2 = 0.42$, $p < 0.001$). We were also able to reproduce the results listed in Spencer et al. (2012) for the years 2009-2010. However, we also found a significant span in the % explained variance between different time periods: The stepwise correlation analysis of two consecutive years (e.g. 1999-2000) showed $R^2$-values ranging from -0.18 (2007 2008) to 0.93 (2009 2010) (see table 5 in revised manuscript). 2005-2006 and 2009-2010 were the only periods for which the quality of the DOC/CDOM correlation was sufficient to compare to our study. From this more detailed investigation of the USGS data, we conclude that, in order to utilize in situ sensors for DOC predictions, it is important to continuously monitor the correlation between DOC and CDOM / FDOM. However, we would like to refrain from further speculating as to why the DOC-CDOM relationship appears spatially and temporally unstable, since we do not have sufficient information to do so. The same holds true

for the differences in FDOM readings between SATURN-08 and SATURN-05. As mentioned in our response to reviewer 1, the sensors at SATURN-05 have since been recalibrated. As a consequence of this comment, we have revised the discussion in section 4.2 and 4.3, pages 13 and 14. We recognize the reviewers call for overlaying our own data with that of USGS. However, a meaningful, direct comparison between data obtained in this study and data collected by USGS is not feasible, due to the small overlap of samples (n = 4).

RC2: In a few places in the introduction and abstract, the text seems to suggest river discharge is one of biogeochemical factors (e.g., p3 at the beginning), which is not. It is hydrology. Need to be consistent throughout the paper.

Response: We deleted "biogeochemical" on page 1, L. 23 and added "and hydrological" on page 3, L.1

RC2: p2, L2, the reference is still from 1981; I believe there should be newer update.

Response: We have replaced the citation with a more recent publication and adjusted the estimated flux accordingly.

RC2: Eq 1. Please specify what is the purpose of calculating napherian absorbance.

Response: The use of napierian absorbance is based on its common usage within the marine CDOM community and is thus intended to present data in a comparable unit.

RC2: p6, L21, what is LOBO?

Response: Land/Ocean Biogeochemical Observatory. We have added an explanation to the manuscript.

RC2: 6. Section 2.6, no performance metrics of the in-situ sensors were given. What are their precision/accuracy, data quality etc.? Need to be careful what the quality these sensors can deliver. They may not always give the data quality as suggested by the manufactures. Should more systematically describe how these sensors were used

and calibrated, and how the data were quality controlled, and in what accuracy and precision these data can be trusted.

We have modified section 2.6 to provide better detail on the sensor maintenance and data processing and included performance metrics provided by the manufacturer in the supplementary material of this comment and the revised manuscript. We agree with the statement that performance as specified by the manufacturer might not be identical to field measurements. However, that is mainly caused by conditions that cannot be recreated in a laboratory environment. In our opinion, regular cleaning and inspection, as well as frequent recalibration by the manufacturer are appropriate to handle this challenge.

RC2: p6, L30, 'No corrections were made to the in-situ nitrate and phosphate measurements'. Why not, since there are discreet samples for nutrients measured?

Response: In situ Nitrate correlated with an R2 of 0.99, thus no correction was necessary. For Phosphate, corrections were not feasible due to the fact that in situ concentrations were mostly below the limit of detection. Explanations were added to the manuscript to clarify the reviewers concern.

RC2: p9, L18, 65%. It is more like 60% to me.

Response: Thanks for pointing this out, the text was corrected to 61 %.

9. p10, L8, 'parametric correlation analysis'. What is this analysis exactly? Linear analysis?

Response: The manuscript text was modified to clearly state that a parametric, linear correlation analysis was performed.

RC2: p10, L8-13. The description here is confusing. Fig. 5 did not show Flu vs. FDOM relationship. What the two relationships here mean or imply then?

Response: We added a figure to the supplement of this response (Supplement Fig. 1)

to show the relationship between in situ and laboratory fluorescence. The reference in the parenthesis was modified accordingly. Moreover, a mistake in the reported R2 was corrected (R2 was 0.93 instead of 0.95). Please also note that we will provide Fig. 1 to the supplement of the revised manuscript.

RC2: Between Section 4 and Section 4.1, the text here does not seem follow a logic way for presentation. May want to organize it into one or few sub-sections.

Response: We have added two subsections that help to emphasize the structure of this part of the discussion.

RC2: p12, L15, says rainfall caused decline in phytoplankton abundance. Any data or figure to show?

Response: This was also pointed out by reviewer 1. We have expanded this conclusion by citing supporting data (significant inverse correlation between chl a and discharge in the respective two week window).

RC2: p12, L26, why there is increased terrestrial nitrate runoff in winter?

Response: Increased terrestrial runoff is likely a consequence of increased winter rainfall in the study area. This explanation was added to the manuscript.

RC2: p12, L33, 'high' rainfall during May 2013. But it says low rainfall before this?

Response: "High amount" was substituted with "frequent" to improve the explanation. The overall amount of rain in the season was low, but the frequent occurrence of rain in May likely led to an increased discharge.

RC2: p14, L8, -7.57 vs. -4.7. I think the measurement error margin was about 10 umol/L, correct? So the difference is within the error margin, is it not? If yes, the conclusion here is not valid. How much of difference is in slopes here? Can that slope difference tell us some information?

Response: The error of DOC measurements should be treated independently of the

intercept uncertainty. The only available metric to judge the significance of the difference between the two intercepts is their respective standard error (SE). Since DOC and CDOM did not correlate well in Spencer et al. (2012), the SE is rather high (3.3) compared to our study (0.7). After reviewing this information, we decided to alter this part of the discussion. The modified version also discusses the slope values. Thank you for raising this point.

RC2: Figs 2 and 4. There are no captions for sub plots (a), (b),...

Response: More detailed explanation on panels was added to both figure captions.

RC2: Figure 10. Is there an explanation why USGS data are different than the modeled DOC?

Response: We do not have definitive explanations for the differences visible in Fig. 10. However, the most likely explanation for the differences is that differences originate from lateral input of organic matter between SATURN-08 and SATURN-05. In our opinion, further explanations are too speculative.

Please also note the supplement to this comment:
http://www.biogeosciences-discuss.net/bg-2016-263/bg-2016-263-AC2-supplement.pdf

**Supplement:**

**Supplement to:**

**Seasonal variability of dissolved organic matter in the Columbia River: In situ sensors elucidate biogeochemical and molecular analyses**

Urban Johannes Wünsch[1,2], Boris Peter Koch[2,1], Matthias Witt[3], Joseph Andrew Needoba[4]

[1] University of Applied Sciences Bremerhaven, An der Karlstadt 8, D-27568 Bremerhaven, Germany
[2] Alfred-Wegener-Institut Helmholtz-Zentrum für Polar- und Meeresforschung, Am Handelshafen 12, D-27570 Bremerhaven, Germany
[3] Bruker Daltonik GmbH, Fahrenheitstraße 4, D-28359 Bremen, Germany
[4] Oregon Health and Science University Institute of Environmental Health, 3181 Southwest Sam Jackson Park Road, Portland, Oregon 97239-3098, United States of America

*Correspondence to*: Urban J. Wünsch (urbw@aqua.dtu.dk)

**Tables**

**Supplement Table 1. Performance specification of in situ sensors used at SATURN-05 and SATURN-08 (provided by the manufacturer). Post-processing was done as specified in the main text. N.A.: Not available, QSDE: Quinine Sulfate Dihydrate equivalent.**

| Parameter | Sensor name | Manufacturer | Accuracy | Detection limit |
|---|---|---|---|---|
| Nitrate | SUNA | Satlantic | $\pm$ 2 µmol L$^{-1}$ | 0.5 µmol L$^{-1}$ |
| Temperature | WQM | SeaBird Coastal | 0.002 °C | N.A. |
| Dissolved Oxygen | WQM | SeaBird Coastal | 2% | N.A. |
| Chlorophyll a | WQM | SeaBird Coastal | 0.2 % | N.A. |
| Turbidity | WQM | SeaBird Coastal | 0.1% | N.A. |
| FDOM | ECO | WET Labs | 0.09 QSDE | N.A. |
| Phosphate | HydroCycle-PO4 Phosphate | Seabird Coastal | 0.15 µmol L$^{-1}$ | 0.075 µmol L$^{-1}$ |

5

**Figures**

[Figure]

**Supplement Figure 1. Comparison between in situ FDOM and FDOM determined by spectrofluorometry measurements of filtered samples ($R^2 = 0.92$).**